# Chromosome 22q11.2 Deletion Syndrome: A Comprehensive Review of Molecular Genetics in the Context of Multidisciplinary Clinical Approach

**DOI:** 10.3390/ijms24098317

**Published:** 2023-05-05

**Authors:** Aleksandra Szczawińska-Popłonyk, Eyal Schwartzmann, Zuzanna Chmara, Antonina Głukowska, Tomasz Krysa, Maksymilian Majchrzycki, Maurycy Olejnicki, Paulina Ostrowska, Joanna Babik

**Affiliations:** 1Department of Pediatric Pneumonology, Allergy and Clinical Immunology, Institute of Pediatrics, Karol Marcinkowski University of Medical Sciences, 60-572 Poznań, Poland; 2Medical Student Scientific Society, English Division, Karol Marcinkowski University of Medical Sciences, 60-572 Poznań, Poland; 3Medical Student Scientific Society, Karol Marcinkowski University of Medical Sciences, 60-572 Poznań, Poland; 4Gynecology and Obstetrics with Pregnancy Pathology Unit, Franciszek Raszeja Municipal Hospital, 60-834 Poznań, Poland

**Keywords:** 22q11.2 deletion, microdeletion, DiGeorge syndrome, velocardiofacial syndrome, dysmorphism, inborn errors of immunity, thymus, congenital heart defect, hypocalcemia

## Abstract

The 22q11.2 deletion syndrome is a multisystemic disorder characterized by a marked variability of phenotypic features, making the diagnosis challenging for clinicians. The wide spectrum of clinical manifestations includes congenital heart defects—most frequently conotruncal cardiac anomalies—thymic hypoplasia and predominating cellular immune deficiency, laryngeal developmental defects, midline anomalies with cleft palate and velar insufficiency, structural airway defects, facial dysmorphism, parathyroid and thyroid gland hormonal dysfunctions, speech delay, developmental delay, and neurocognitive and psychiatric disorders. Significant progress has been made in understanding the complex molecular genetic etiology of 22q11.2 deletion syndrome underpinning the heterogeneity of clinical manifestations. The deletion is caused by chromosomal rearrangements in meiosis and is mediated by non-allelic homologous recombination events between low copy repeats or segmental duplications in the 22q11.2 region. A range of genetic modifiers and environmental factors, as well as the impact of hemizygosity on the remaining allele, contribute to the intricate genotype-phenotype relationships. This comprehensive review has been aimed at highlighting the molecular genetic background of 22q11.2 deletion syndrome in correlation with a clinical multidisciplinary approach.

## 1. Introduction

The chromosome 22q11.2 deletion syndrome (22q11.2 DS), also known as DiGeorge syndrome (DGS) or velocardiofacial syndrome (VCFS), is a genetic condition resulting from the impaired development of structures originating from the third and fourth pharyngeal pouches in the germinal stage. The clinical features of the syndrome include hypoparathyroidism and hypocalcemia, thymic hypoplasia, conotruncal heart defects, facial dysmorphism, and palatoschisis. The complex phenotype of children affected with 22q11.2 DS may show considerable intersubject variability, and the expanded clinical manifestations comprise craniofacial, neurological, cognitive, behavioral, ocular, speech and hearing, musculoskeletal, and internal organs, as well as airway, gastrointestinal or renal abnormalities [1]. The phenotypes may vary considerably among patients with immunodeficiency and immune dysregulation, including autoimmunity, allergy, and lymphoproliferative sequelae.

The incidence of 22q11.2 DS has been estimated to range from 1:3000 [2] to 1:4000 live births [1], placing it among frequent syndromic diseases. However, the rate of its clinical suspicion is still challenging, and the process of establishing the definitive diagnosis is an odyssey [3] due to the remarkable heterogeneity of clinical phenotypic expressions and overlapping manifestations with other categories of syndromic disorders [4,5,6]. The disease entities, such as, but not limited to, coloboma-heart defects-choanal atresia-retardation of growth and development-ear anomalies (CHARGE) syndrome [7,8], cardio-facio-cutaneous (CFC) syndrome [9,10], and Takenouchi-Kosaki syndrome (TKS) [11,12], require special pediatrician awareness and multidisciplinary care, as their distinctive phenotypes are associated with immunodeficiencies. Furthermore, a common denominator of 22q11.2 DS and other syndromic disorders is an increased susceptibility to recurrent infections, which may not result only from inborn errors of immunity; a multiplicity of developmental anatomical malformations and organ dysfunctions are also important contributing factors. These extra-immune phenotypes in 22q11.2 DS, with neurological, psychomotor, hormonal, circulatory, and respiratory pathophysiological mechanisms, are substantially underpinning the definitive clinical manifestations.

However, there is a paucity of reports on the clinical course of 22q11.2 DS in pediatric patients, and the data are scattered, thereby making them difficult to apply in the all-embracing pediatric practice. In this review, we aimed to establish a comprehensive molecular genetic and clinical approach to 22q11.2 DS addressed to pediatricians and specialists in other fields of medicine who encounter children affected with this syndrome and provide in-depth clinical care. We also sought to determine what is known and what is new in the multidisciplinary genetic and clinical diagnostic process and prophylactic measures, as well as therapeutic modalities in 22q11.2 DS.

## 2. Pathophysiology

Several clinical disorders make up 22q11.2 DS, such as velocardiofacial (VCF, pharyngeal dysfunction, cardiac anomaly, facial dysmorphism) and DiGeorge (DGS, cardiac anomaly, hypoparathyroidism, thymic hypoplasia) syndromes, a group of disorders described by the acronym CATCH22 (cardiac defect, abnormal facial features, thymic hypoplasia, cleft palate, hypocalcemia). The nomenclature may be confusing due to the remarkable heterogeneity of clinical phenotype and marked intersubject variability of the presenting symptomatology in 22q11.2 DS [13]. A variety of anatomical and functional developmental abnormalities occur in the fetus and multiple conditions appear later in childhood. The syndrome is characterized by haploinsufficiency resulting from a hemizygous deletion in the region 11.2 on the long arm of chromosome 22, meaning that the gene alleles have no homologic counterparts. Interestingly, a deletion on 22q11.2, in combination with a single gene variant on the other allele, can reveal autosomal recessive conditions, such as Bernard-Soulier syndrome with thrombocytopenia and increased megakaryocytes (platelet glycoprotein 1B beta-polypeptide, *GP1BB* variant) and cerebral dysgenesis (polymicrogyria), neuropathy, ichthyosis, and palmoplantar keratoderma (CEDNIK) syndrome (synaptosomal-associated protein 29 kDa, *SNAP29* variant) [14].

The deletion types and sizes in 22q11.2 DS show a high degree of variability due to several low-copy-number repeat sequences (LCR22A, LCR22B, LCR22C, LCR22D, LCR22E, and LCR22F) flanking the deleted region. Most patients (about 90%) show a 2.54 Mb heterozygous deletion comprising four repeats extending from LCR22A to LCR22D and involving approximately 40 genes [15]. The disease-causing genes mapping to the region spanned by LCR22A-LCR22D are, among others, *TBX1*, proline dehydrogenase *PRODH*, catechol-O-methyltransferase *COMT*, cell division cycle 45 *CDC45*, *GP1BB*, *SNAP29*, and DiGeorge critical regions *DGCR2*, *DGCR6*, *DGCR6L*, and *DGCR8*. A proximal 1.25 Mb deletion comprising LCR22A to LCR22B affects about 5% of individuals with 22q11.2 DS, whereas 2% of them show a deletion spanning the LCR22A to LCR22C region, and the next 5% have a smaller, atypical, nested deletion extending from LCR22B or LCR22C to LCR22D. Distal deletions, flanked by LCR22D-E and LCR22D-F, have been reported less frequently [15,16]. The complexity of 22q11.2 genetics is even more prominent due to the regulatory effect of deleted *DGCR6*, *DGCR6L*, and *DGCR8* gene alleles and mi-RNAs, such as miR-185, miR-4716, miR-3618, miR-1286, miR-1306, and miR-6816. Several miRs—miR-96-5, miR-451a, and miR-17-92—target *TBX1*, thereby influencing its expression and, consequently, the clinical phenotype [17,18,19]. The 22q11.2 region encompassing protein-coding genes, long non-coding RNAs, microRNAs, and genes expressed in functional haploinsufficiency in 22q11.2 DS are displayed in Figure 1. Additionally, genome-wide DNA methylation analysis in a group of patients with 22q11.2 DS has demonstrated differentially methylated regions located apart from the chromosome 22, which may involve clinically relevant genes or regulatory elements, thereby determining the variability of phenotypic features. This phenomenon may also shed light on the impact of 22q11.2 haploinsufficiency in altering the genomic methylation level [20].

The de novo occurrence of the syndrome is observable in about 90–95% of affected individuals in all populations, yet autosomal dominant inheritance has also been ascertained in 6–28% of subjects [17]. With its prevalence ranging broadly from 1:2000 to 1:6000 live births [14], 22q11.2 DS is put among frequent syndromic disorders. In seemingly anatomically normal fetuses, 22q11.2 deletion occurs with an approximate frequency of 1:1000, and it reaches 1:100 in fetuses with major structural defects, such as congenital heart disorders [14]. Furthermore, the current estimation of 22q11.2 DS frequency needs to take into consideration the reproductive fitness and increase in survival of children who can pass on the deletion to their offspring, thereby increasing the frequency of its occurrence [21,22,23]. Although 22q11.2 DS is the most common microdeletion in humans, it may be underdiagnosed in diverse populations due to the heterogeneity of clinical phenotypic features, including less recognizable facial dysmorphism, largely depending on the patient’s ethnicity [24]. Thus, the recognition rate of 22q11.2 in affected children of non-European—in particular, African or Asian descent—may be insufficient and the actual prevalence underestimated.

## 3. The Burden of 22q11.2 in Pediatrics

With its average frequency estimated to be 1:4000 live births and 1:1000 in unselected fetuses [14], 22q.11.2 DS has a global burden and remarkable impact on affected patients, their families, society, and health providers, as well. Whereas the number of patients reported in the medical literature is disproportionately lower than anticipated due to the estimated frequency, important concerns are raised regarding all the newborns awaiting the genetic molecular diagnosis or children who escaped from the timely recognition of this syndrome [25]. The latter group of undiagnosed individuals may embrace all those children who experience a protracted period of the diagnostic odyssey [2]; they are consequently burdened by unfavorable outcomes. Of note, 22q11.2 DS is an important cause of morbidity and mortality across the lifespan, and both major birth defects and complications that develop later in childhood or even adulthood make an early definitive diagnosis of paramount importance. In addition to being the most common cause of syndromic palatal anomalies and velopharyngeal dysfunction [26,27], 22q11.2 DS is the also most common cause of schizophrenia [28,29] and the second most common cause of developmental delay and congenital heart disease after Down syndrome, as well as a more common cause of conotruncal heart disease [30,31]. The occurrence of 22q11.2 deletion is found in about 2% of all subjects with congenital heart disease [32,33], and, thus, the syndrome should always be considered in prenatal settings in fetuses affected with congenital heart developmental abnormalities [34]. Notably, after the surgery for congenital heart disease, the rate of postoperative non-cardiac complications, such as infections or the need for dialysis [35], length of stay in the intensive care unit, and hypocalcemia [36], is higher in children with 22q11.2 DS than in those with no chromosomal defects. Furthermore, severe forms of congenital heart disease are a primary factor contributing to a lower life expectancy in adults with 22q11.2 DS because the probability to survive to the age of 45 is approximately 72% in those individuals [22].

## 4. Clinical Symptomatology

Craniofacial dysmorphism in 22q11.2 DS is characterized by a multiplicity of facial phenotypic features. Some of these facial characteristics are not easily recognized clinically in children as they are mild in nature, and age is an important determinant of facial morphology. These features include an elongated face, hypertelorism, wide nasal bridge, hooded eyelids, upslanted palpebral fissures, epicanthus, long nose with a bulbous tip, narrow alar base, short philtrum, small mouth, micrognathia, and low-set, posteriorly rotated, small ears. The retrognathic posture of the mandible, inward displacement of the lower part of the face and the ocular region, and an enlarged cranial base angle have been identified as the uppermost dysmorphology features in the cephalometric and spatially dense 3-dimensional analysis [37]. High prevalence of dental features, including tooth abnormalities of their eruption, shape, and number—such as agenesis of permanent dentition or supernumerary teeth, as well as enamel hypomineralization and hypoplasia—are attributable to 22q11 DS [37,38,39].

Among oropharyngeal developmental disorders, palatoschisis is a frequently observable phenotypic feature, occurring in the form of open cleft, submucosal cleft, palatopharyngeal disproportion, and cleft lingula [40,41]. Velopharyngeal insufficiency and hypotonia accompanied by hypernasal speech have been estimated to occur in as many as 30–80% of children with 22q11.2 DS, rendering the surgical intervention particularly challenging due to the complexity of coexisting serious medical conditions [41,42]. Over the past decade, the number of clinical centers of excellence dedicated to the care of 22q11.2 DS patients, with surgeons specialized in velopharyngeal operations, has grown across different regions of the world.

Developmental disorders of the upper airways in 22q11.2 DS include congenital laryngeal anomalies. Laryngeal atresia due to a glottic web or partial laryngeal stenosis, laryngomalacia, laryngeal cleft, and vocal fold abnormalities are characteristic parts of the syndromic features [43,44]. The laryngeal web has been estimated to be the most frequent upper airway anomaly and 40% sensitive for suggesting a diagnosis of 22q11.2 DS [45]. Velopharyngeal and laryngeal anomalies may be accompanied by lower respiratory tract disorders, such as tracheal stenosis, tracheo- and bronchomalacia, short trachea with reduced numbers of tracheal rings, and tracheoesophageal fistula, as well as aberrant tracheal bronchus [46,47]. These structural airway abnormalities may be related to obstruction due to anatomically and functionally reduced patency, recurrent and persistent respiratory tract infections, atelectasis of the lung, and pulmonary edema resulting from congenital heart disease. All these overlapping clinical encumbrances may be significant causes of morbidity and mortality in children with 22q11.2 DS and result in respiratory distress and the need for ventilation support and tracheostomy. Impaired coordination of swallowing and breathing, the risk of aspiration, and nasopharyngeal reflux, as well as the need to provide caloric and nutritional support, necessitate the placement of an enteral percutaneous feeding tube [48].

Congenital heart disease has been recognized in approximately 60–80% of children affected with 22q11.2 DS. The most commonly occurring subset of cardiac anomaly is a conotruncal defect, such as tetralogy of Fallot, pulmonary atresia with ventricular septal defect, truncus arteriosus, interrupted aortic arch type B, conoventricular and/or atrial septal defects, and aortic arch anomalies [32]. The anatomical complexity of congenital heart disease in 22q11.2 DS requires special perioperative care, particularly in children with pulmonary atresia-ventricular septal defect (PA-VSD) and major portopulmonary collateral arteries (MAPCAs), which pose the risk of increased mortality due to vasomotor instability, airway hyperresponsiveness, coexisting airway abnormalities, increased frequency of airway bleeding, and infectious—mainly fungal—complications [49]. The syndrome is therefore associated with unfavorable early perioperative results which, in turn, may be associated with worse cognitive and neuropsychiatric outcomes [50]. Cardiovascular complications are the most common cause of premature death in adults with 22q11.2 DS. Major associated conditions, such as hypocalcemia, thyroid disorders, autoimmune diseases, behavioral problems, and neurodevelopmental disability contribute to the worse fallout [51]. Interestingly, it has been suggested that variance in congenital heart disease, conotruncal-type penetration, in the population of individuals with 22q11.2 DS may be associated with variants in GH22J020947 affecting the expression of *CRKL* encoding for a cytoplasmic adaptor protein (CRK-like proto-oncogene) involved in growth factor signaling [52]. Moreover, it has been hypothesized that rare copy number variants outside the deleted 22q11.2 region may act as modifiers of the risk of congenital heart disease in 22q11.2 microdeletion-affected children [53].

The complex and variable phenotypes related to 22q11.2 DS also include mild to moderate intellectual disability, which occurs in approximately one-third of affected pediatric patients [54]. In these patients, global reduction in brain volume, with widespread decay in frontal, temporal, parietal, and occipital lobes, atrophy of the cerebellum and hippocampus, as well as polymicrogyria and altered cortical thickness are observable in neuroradiological imaging [55,56]. The syndrome is associated with neurological disorders, such as epilepsy and increased incidence of seizures, which may be provoked by other concomitant conditions and complications, including hypoxia due to congenital heart disease and cardiac surgery, hypocalcemia, thyroid hormonal dysfunction, infections, and fever [57]. Movement disorders in 22q11.2 DS include catatonia and the increased risk of Parkinson’s disease [58].

Noteworthy, 22q11.2 DS is one of the rare examples of a cytogenetic abnormality occurring in conjunction with a psychiatric disease, schizophrenia, autism spectrum disorders (ASD) with some level of social-communication impairment, and attention deficit/hyperreactivity disorder (ADHD), which occur in as many as 20% of affected children and adolescents [54,59]. The spectrum of psychopathology in 22q11.2 DS is complex and comprises anxiety disorders, disruptive, and mood disorders. Importantly, symptoms of psychotic conditions may be misinterpreted and wrongly attributed to cognitive impairment, with low social and communicative skills. Moreover, both low intellectual ability early in life and its subsequent decline, impaired verbal and perceptual reasoning abilities, as well as altered auditory and visual processing from preschool to adolescence, are associated with increased risk of schizophrenia [60,61,62,63]. Ocular abnormalities are frequently encountered in children with 22q11.2 DS in the form of refractive errors, strabismus, amblyopia, and other structural ophthalmological disorders. such as posterior embryotoxon aka Axenfeld-Rieger anomaly, which may contribute to learning difficulties and cognitive impairment [64,65].

Endocrine manifestations are hallmarks of 22q11.2 DS and primarily comprise hypoparathyroidism, thyroid dysfunction, and growth retardation [66,67]. Hypocalcemia is considered a classical feature of the syndrome, resulting from low parathormone serum levels, accompanied by 25(OH)D3 deficiency. In addition to the major effect of parathyroid gland endocrinopathy, hypothyroidism and hypomagnesemia may play a role in hypocalcemia by suppressing parathyroid hormone and, thereby, are further contributing factors to hypocalcemia [68]. The presentation of hypocalcemia is more likely to occur at a young age and may vary considerably, ranging from transient neonatal silent hypocalcemia to hypocalcemic tetany and overt hypoparathyroidism throughout the lifespan. Affected patients may also experience fatigue and paresthesia, as well as more severe manifestations including low seizure threshold and prolongation of the QT interval, which significantly affect both early life neurodevelopment [69] and clinical outcome in cardiac and non-cardiac surgical procedures [70,71,72]. Thyroid dysfunction often presents later, in older childhood or adulthood, and is commonly due to autoimmune thyroiditis. Thyroid autoantibodies, particularly anti-thyroperoxidase antibodies, have been found in up to 5% of children and in approximately 30% of adults affected with 22q11.2 DS [73]. Two clinical disease entities of autoimmune thyroiditis have been reported in children with the syndrome: Hashimoto thyroiditis occurs in 20% of them and is characterized by inhomogeneous thyroid echostructure and progression from normal function to hypothyroidism and, less frequently, Grave’s disease, which presents with overt hyperthyroidism and is observable in 1.4% of affected pediatric patients [67,73,74,75,76,77]. Congenital thyroid gland abnormalities have also been recognized in children with 22q11.2 DS and include an absent thyroid isthmus, retrocarotid and retroesophageal extension, and absence/hypoplasia of the left thyroid lobe. It is noteworthy that developmental thyroid disorders coexist in as many as 71% of children with congenital heart disease, compared to 31% of those with normal thyroid volume, highlighting the role of the *TBX1* gene in the formation of cardiac outflow structure and positioning of the thyroid gland [78]. Growth restriction due to a growth hormone deficiency that is consistent from infancy to final height has been reported, and the association between patients’ height and congenital heart disease has been shown in pediatric and young adult patients with 22q11.2 DS [66]. Whereas immune dysregulation is associated with 22q11.2 DS, autoimmune endocrinopathies, such as diabetes mellitus with serum insulin antibodies [79] or adrenal antibodies with normal adrenal function [80], are concomitant clinical features. Metabolic disorders, such as type 2 diabetes [81], hypertriglyceridemia [82], and obesity [83], have been noted in patients with the syndrome, with the high frequency of the latter condition reaching approximately 43% of adult individuals affected with 22q11.2 DS [83].

The summary of multisystemic clinical phenotypic features and their estimated frequency in patients with 22q11.2 deletion syndrome, based on literature reviews, is shown in Table 1 [13,14,32,37,38,39,43,44,46,47,55,56,64,65,66,67,84].

## 5. Immune Deficiency and Immune Dysregulation

Immunodeficiency is a key feature of 22q11.2 DS and is secondary to thymic aplasia or hypoplasia with subsequent impaired thymocyte development. The third and fourth pharyngeal pouches are a common embryonic precursor for the thymus, parathyroid glands, and conotruncal regions of the heart. In 22q11.2 DS, maldevelopment of these organs is due to impaired migration of the neural crest cell into pouch ectoderm [85]. In the setting of abnormal thymic migration, but with the preservation of residual microscopic nests of thymic epithelial cells, mild to moderate reductions in T cell numbers accompanied by only a mild deficit in T cell function occur in most affected children. However, even in these patients, the T-cell thymic output of recent thymic emigrants, assessed by T-cell receptor excision circles (TRECs) analysis, is very low and decreases with age [86,87]. Full thymic aplasia appears occasionally, in approximately 1% of cases with 22q11.2 DS [86,88]. Interestingly, beyond the *TBX1* hemizygosity, other genes, such as *CRKL* in the affected 22q11.2 region, may also have a gene dosing, modifying effect on the phenotypic expression of the syndrome. CRKL is expressed in neural crest-derived tissues and involves thymic development. The effect of compound heterozygosity for TBX1 and CRKL deletion on clinical features and thymus development is additive [89].

Consequently, a wide spectrum of T-cell alterations is seen in 22q11.2 DS, ranging from near normal to near completely immunodeficient. Mild T cell immunodeficiency may be found in children with apparently hypoplastic thymus because ectopic retropharyngeal thymic tissue may be preserved [90,91]. Furthermore, dynamic changes in immunodeficiency are observable over time, and the direct effect of thymic hypoplasia on T cell counts is most apparent in early infancy [86,87]. They tend to normalize by adulthood in most patients, due to the increased secretion of interleukin (IL)-7 that stimulates the thymic output and peripheral proliferation of T cells [92]. The most common deficits include low total CD3+ T cell percentage and absolute count, as well as low numbers of naive CD4+ T helper and CD8+ T cytotoxic/suppressor cells. The naive T cell compartment shows a progressive decline with patients’ age, leading to a predominantly memory phenotype of T cells at the periphery [93].

Referring to anatomical maldevelopment and dysfunction in generating T cells, the humoral immunodeficiency and B cell abnormalities in children with 22q11.2 DS are secondary to T cell deficits [94]. Low immunoglobulin production, most frequently affecting IgM and occasionally IgG and rendering the need for immunoglobulin replacement therapy, as well as the defective immune response to polysaccharide antigens, have been reported in children with the syndrome [95]. Among the B cell subsets, low switched memory B cells expanding by adulthood, accompanied by decreased somatic hypermutation despite increased follicular T helper cells, have been shown, reflecting a dysregulated B cell compartment and compromised T cell help [96,97,98]. Recurrent respiratory infections, such as adenotonsillitis, otitis media, bronchitis, pneumonia, as well as sepsis [99,100], also occur.

Immune dysregulation in 22q11.2 DS has been foremostly ascribed to T-cell lymphopenia and deficiency in CD3+CD4+CD25++, FOXP3+ regulatory T cells, which play a crucial role in maintaining immune homeostasis and self-tolerance. Reduced thymic output related to an increased naive T cell subset, and subsequent T regulatory cell activation, control the expansion of the T cell compartment. A reduced number of regulatory T cells, with their activated phenotype and loss of suppressive capacity in children with 22q11.2 DS, are key features of deregulated T cell homeostasis [101]. Beyond the T cell compartment, immunophenotype anomalies also encompass peculiar B cell developmental disorders with increased naive B cells and deficit in switched memory B cells [102] which are biomarkers of immune dysregulation in 22q11.2 DS. It has been postulated that the individual patient’s immunophenotype may be influenced by genetic modifiers outside the microdeletion locus which regulatethe expression of *TBX1*. Rare DNA variants in transcriptional regulators involved in retinoic acid signaling, *NCOR2* and *EP300,* were found to be associated with parameters of the immune functions, such as immunoglobulin levels, lymphocyte response to antigens and mitogens, and flow cytometric lymphocyte compartment. Retinoic acid plays an important role in maintaining immune homeostasis by enhancing the differentiation of regulatory T cells, modulating epithelial and mucosal immune responses, and regulating proinflammatory cytokine activity. Hence, genetic modifiers contributing to the individual’s genetic background and modulating variable penetrance may influence the immune response in 22q11.2 DS [103,104].

Autoimmune disorders have been described in as many as 23% of pediatric patients with the syndrome [105], manifesting as autoimmune thyroid disease, juvenile idiopathic arthritis, autoimmune cytopenia (thrombocytopenia, hemolytic anemia, neutropenia), celiac disease, psoriasis, vitiligo, autoimmune hepatitis, and inflammatory bowel disease [100,105,106,107]. It has also been hypothesized that psychotic disorders, developmental regression, and cognitive impairment in children with 22q11.2 DS may have a causal relationship with autoimmune encephalitis [108].

The peripheral homeostatic expansion of T cells driven by low thymic output and T cell lymphopenia may contribute to Th2-skewed lymphocyte phenotype and atopic manifestations, such as eczema and asthma [109]. In these children with 22q.11.2 DS, the overall frequency of atopic diseases has been estimated to reach 70% and the frequency of asthma to as many as 50% [110]. Coexisting gastroesophageal reflux and sinopulmonary infections may have an impact on its clinical course [111].

The thymus dysfunction and immunophenotypic abnormalities within the T cell compartments in 22q11.2 DS make affected children susceptible to cancerogenic viruses, such as Epstein-Barr virus (EBV) and human papilloma virus (HPV), that might be linked to an increased risk of malignant transformation. It has also been postulated that chronic immune activation of peripherally expanded T cell population in dysfunctional cellular immunity may underpin the predisposition to developing lymphoproliferative disorders and lymphomagenesis [112], as T cell and B cell lymphomas [113,114,115], as well as acute lymphoblastic leukemia [116], have been reported in children with 22q11.2 DS. The spectrum of malignancies in affected pediatric patients also includes solid tumors—namely Wilms tumor—hepatoblastoma, neuroblastoma, thyroid carcinoma [116], pineoblastoma [112], and xanthoastrocytoma [117].

## 6. Diagnosis

The chromosomal 22q11.2 DS is a clinically highly variable microdeletion syndrome with differently expressed phenotypes, with wide interfamilial and intrafamilial variability in patients sharing the same genetic underpinnings [118]. This is due to both the remarkable complexity of the 22q11.2 region with LCR blocks and the high susceptibility of this region to meiotic errors, as well as the epigenomic and environmental factors influencing the phenotypic variability [119]. Based on functional genomic assessments, it has also been hypothesized that theories on single-gene haploinsufficiency in 22q11.2 DS cannot be supported. In the setting of diminished 22q11.2 gene dosage, shared molecular functions, convergence on cellular processes, and related consequences on the genetic level point to the matrix or multigenic interactions that translate into the multiplicity of phenotypes [120]. All these genetic factors, together with age-related developing symptomatology, contribute to diagnostic challenges in 22q11.2 DS pediatric patients on the diverse individual, family, social, and population levels. The summary of modifying variants influencing the *TBX1* penetrance and related phenotypic expression is shown in Table 2 [17,18,19,52,53,103,104,118,120,121,122,123,124].

The diagnosis of 22q11.2 DS has been traditionally based on the recognition of clinical features and cytogenetic testing using the fluorescence in situ hybridization (FISH) technique. FISH is perceived as the golden standard genetic testing method to confirm the diagnosis of microdeletion syndromes. However, poor clinical accuracy, the low confirmatory rate in the screening of suspected microdeletion syndromes, and failure to detect other than the targeted microdeletion are the major drawbacks of this method. An important limitation of this method is a failure to identify atypical and nested deletions because it can recognize deletions in the proximal part of the critical region, including the typical LCR22A-D deletion. FISH is admittedly inexpensive, yet still a highly labor-intensive and time-consuming procedure [125]. Importantly, due to a marked clinical variability from minimal to full manifestation in patients with 22q11.2 DS, precision in defining clinical criteria is important for referring patients to undergo FISH analysis. Since FISH alone cannot provide a reliable diagnosis of 22q11.2 DS, other diagnostic methods have been developed, such as comparative genomic hybridization (CGH), multiplex ligation-dependent probe amplification (MLPA), multiplex quantitative real-time polymerase chain reaction (qPCR), and high-resolution single-nucleotide polymorphism (SNP) microarray analysis [126]. Although the FISH method is still routinely used in laboratories, the MLPA assay has been perceived as an alternative that is superior to the FISH technique as it is less costly, less time-consuming, and laborious, and does not require cell cultures. It has been proposed that a locus-specific approach, using FISH or MLPA assays, could be offered to children strongly meeting clinical and dysmorphology criteria for 22q11.2 DS [126,127]. Even though FISH is a state-of-the-art procedure for patients with the clinical suspicion of 22q11.2 DS, at present, patients are usually diagnosed by indirect whole genome studies [128]. Referring to the regulatory role of the *TBX1* gene during development of the heart, thymus, and parathyroid glands, as well as during formation of the palate, teeth, and craniofacial features, there has been growing evidence that *TBX1* is a candidate gene for 22q11.2 DS [129,130]. Therefore, in patients with clinically evident disease in whom a deletion of 22q11.2 has not been identified by FISH or microarray tests, TBX1 gene testing is recommended. Genotype first approach has therefore been proposed, and whole genome sequencing as the first-line method in the not-too-distant future, with FISH to be used as a confirmation of patient and family screening results [125].

Newborn screening (NBS) for severe combined immunodeficiency (SCID) is identifying a subset of infants with 22q11.2 DS due to T cell lymphopenia and low TREC numbers, and, hence, it is not a universally reliable detecting method. Therefore, direct NBS for 22q11.2 DS to recognize affected neonates using the genomic approach with multiplex qPCR assay, targeted to the *TBX1* gene within the LCR22A-B region and *CRKL* within the LCR22C-D region, has been elaborated [131].

Whereas postnatal phenotypes have been widely characterized and categorized, prenatal diagnosis of 22q11.2 DS remains challenging due to a low rate of inheritance (10% of cases) and mild unrecognized parental features. Fetal ultrasound imaging may provide information on findings characteristic for 22q11.2 DS, such as polyhydramnios, hypoplasia or aplasia of the fetal thymus, central nervous system anomalies, such as asymmetric ventriculomegaly and dilated cavum septum pellucidum, and, rarely, skeletal anomalies, among others, such as bilateral talipes and anomalous vertebrae [132].

The golden standard method for detecting 22q11.2 microdeletions remains first trimester screening by chromosomal microarray (CMA), which is performed in invasively obtained prenatal samples, such as chorionic villi and amniotic fluid. An important technical advance in prenatal noninvasive screening for 22q11.2 is cell-free DNA testing [133]. Cell-free DNA fragments present in maternal plasma, which derive from both the mother and the embryo as a result of apoptosis of the cytotrophoblast. An external layer of the placenta is used for qualitative and quantitative assays. The fetal fraction is screened for common trisomy and 22q11.2 deletion, as well as for other rare trisomies and microdeletions. Targeted technologies, such as single-nucleotide polymorphism (SNP)-based, digital analysis of selected regions (DANSR), and targeted capture enrichment assay (TCEA) technologies, as well as the genome-wide methodology massively parallel shotgun sequencing (MPSS), are advanced techniques used in cell-free DNA analysis [133,134,135,136,137].

## 7. Therapeutic Approach

The marked phenotypic variability of 22q11.2 DS is accompanied by a wide scope of immune deficits, ranging from mild to moderate T cell lymphopenia in the partial form of DiGeorge syndrome (pDGS) to profound combined T and B cell immunodeficiency in the complete form (cDGS). In the first case, hypoplastic ectopic thymus or microscopic thymic rests are found and successful spontaneous immunocorrection has been reported, whereas the latter case is characterized by complete athymia [138]. In those most severely immunocompromised children, immune reconstitution may be achieved by thymus transplantation, providing the ability to produce naive T cells showing a broad T cell receptor repertoire. To facilitate the optimal establishment of thymic allograft, stability of comorbidities, such as attaining cardiopulmonary function, upper airway stabilization, appropriate weight gain, and metabolic compensation, are essential to avoid graft failure. The management of concurrent disorders plays a fundamental role in the timing of thymus transplantation, which requires optimal planning and sequencing [139]. Another approach to cDGS is T cell-replete hematopoietic stem cell transplantation; however, due to the absence of the thymus, engraftment of post-thymic T cells may result in poor quality of immune reconstitution [94,140,141]. Although 22q11.2 DS has been perceived as a T cell deficiency, disorders of B cell maturation and reduced numbers and functions in naive, unswitched, and switched memory B cells have also been reported. Humoral immunodeficiency in children with 22q11.2 DS may considerably vary from hypogammaglobulinemia with low all immunoglobulin isotypes and the need to receive immunoglobulin replacement therapy (approximately 6% and 3% of them, respectively) [95] to low serum IgA or IgM and impaired antigen-specific vaccine response in sporadic cases [142,143]. Despite the vast majority of children with 22q11.2 DS having normal serum immunoglobulin levels, due to the variable degree of cellular immunity impairment, they are susceptible to acute and recurrent infections, among others such as sinusitis, otitis media, mastoiditis, pneumonia, urinary tract infections, and viral infections [144,145]. Important questions are then raised about indications for preventive measures against infections in those children who do not qualify for immunoglobulin replacement therapy but present with T-cell lymphopenia. Antibiotic prophylaxis is indicated for children with 22q11.2 DS, first of all in those with low IgA serum levels and panhypogammaglobulinemia presenting with recurrent respiratory tract infections during epidemic season. The prophylactic regimens include daily or alternate-daily use of amoxicillin or azithromycin and co-trimoxazole in children with advanced T-cell lymphopenia posing the risk of Pneumocystis jiroveci infection [136,146,147].

Active immunization in children with 22q11.2 DS requires optimizing to provide vaccination coverage against vaccine-preventable infections. Live vaccine practices with Bacille Calmette-Guerin (BCG), and vaccines against measles-mumps-rubella (MMR), varicella (VAR), and an intranasal live attenuated influenza vaccine (LAIV), are contraindicated in children with this syndrome as T cell lymphopenia makes them susceptible to adverse effects following live immunization (AEFLI). Furthermore, in those children who, due to hypogammaglobulinemia, receive immunoglobulin replacement therapy either intravenously or subcutaneously, live vaccines are inactivated by administered antibodies and thereby are contraindicated. Inactivated vaccines can be safely administered to immunodeficient patients as they do not pose the risk of an uncontrolled spreading of vaccine microorganisms in the patient’s body, and they are not inactivated by supplemented immunoglobulins. However, the immune response to vaccines with antigen-specific antibodies and memory B cell generation may be significantly reduced [148].

However, many individuals with 22q11.2 DS with mild to moderate immunosuppression receive live viral MMR and varicella vaccines despite the known diagnosis and tolerate them well, without serious adverse effects [149,150]. Given the risk of natural infection, the benefits of protection following immunizations with live vaccines outweigh the risks of potential AEFLI. To assess the safety of live attenuated vaccines and evaluate the ability to generate an effective immune response in children with 22q11.2 DS, immunological investigations prior to the administration of live vaccines have been proposed [151]. The recommended immunology workup practices include lymphocyte immunophenotyping with the evaluation of total CD3+ T cells, CD4+ T helper cells, CD8+ T cytotoxic cells, and CD3+CD4+CD45RA+CD31+ recent thymic emigrants, as well as response to mitogen phytohemagglutinin (PHA). Live vaccines can be safely administered in children showing a total T cell count above 0.5 × 10^9^/L, a cytotoxic T cell count above 0.2 × 10^9^/L, and a normal response to mitogen [151].

## 8. Conclusions

Reports on nationwide studies capturing patients with 22q11.2 DS [89,145,152] stand in contrast to the frequency of the syndrome, which has been estimated to occur in approximately from 1:3000 to 1:4000 births and show a remarkable discrepancy between the estimated frequency and the diagnostic rate in 22q11.2 DS. Whereas clinical symptomatology of the syndrome may be heterogeneous and finding the genetic etiology may be arduous due to complex molecular genetics, the role of genetic modifiers, and epigenetic and environmental factors, as well as the influence of mutations on the remaining genes uncovering rare recessive conditions as shown in Table 3 [14,153,154], the clinical diagnosis may be challenging for clinicians. Increased awareness of pediatricians and specialists in different fields of medicine about the broad spectrum of phenotypic features of 22q11.2 DS they may encounter is therefore indispensable. Comprehensive multidisciplinary care should be provided to patients with 22q11.2 DS by cardiologists, cardiosurgeons, endocrinologists, laryngologists, neurologists, surgeons, and geneticists, under the clinical immunologist’s supervision [155,156,157]. Clinical centers of excellence with multidisciplinary expertise in comprehensive care for patients with 22q11.2 DS provide careful monitoring and timely interventions. Multidisciplinary care is associated with significantly higher guidelines adherence and optimal outcomes in affected individuals [158,159,160].

## Figures and Tables

**Figure 1 ijms-24-08317-f001:**
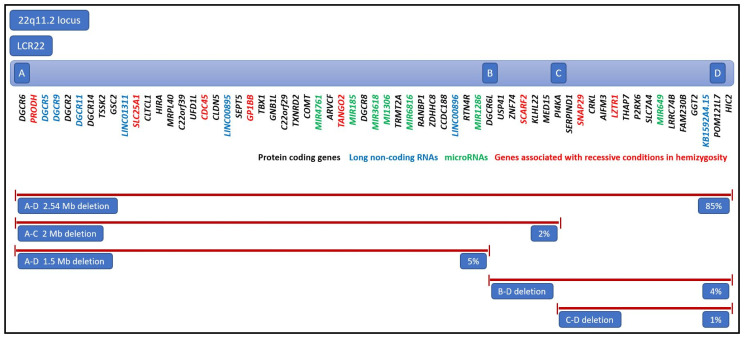
The schematic representation of the 22q11.2 region comprising LCR22 A, B, C, and D sequences and encompassing protein-coding genes, long non-coding RNAs, microRNAs, and genes expressed in functional haploinsufficiency in 22q11.2 DS.

**Table 1 ijms-24-08317-t001:** Multisystemic anatomical anomalies and dysfunctions in patients with 22q11.2 deletion syndrome.

Systemic Involvement	Phenotypic Features	Frequency in 22q11.2 DS
Facial dysmorphism	Elongated face Hooded eyelids Upslanted palpebral fissures Epicanthus Wide nasal bridge Long nose with a bulbous tip Narrow alar base Short philtrum Small mouth Micrognathia and retrognathia Low-set small ears	80–99%
Ocular findings	Posterior embryotoxon Tortous retinal vessels Refractive errors Strabismus Amblyopia	7–70%
Dentition	Delayed teeth eruption Agenesis of permanent dentition Supernumerary teeth Enamel hypoplasia Impaired enamel calcification	2.5%
Palatal anomalies	Velopharyngeal insufficiency and hypotonia Cleft palate Submucous cleft palate Bifid uvula	69–100%
Laryngeal anomalies	Glottic web Laryngeal stenosis Laryngeal cleft Laryngomalacia Vocal fold anomalies	25–43%
Lower airway anomalies	Tracheo and bronchomalacia Tracheal stenosis Short trachea with reduced tracheal rings Aberrant tracheal bronchus Tracheoesophageal fistula	21%
Cardiovascular anomalies	Interrupted aortic arch type B Truncus arteriosus Tetrealogy of Fallot Conoventricular septal defect Isolated aortic arch anomaly Double outlet right ventricle Transposision of the great arteries Hypoplastic left heart syndrome Valvar pulmonary stenosis	49–83%
Genitourinary anomalies	Renal agenesis Multicystic dysplastic kidney Hydronephrosis Duplicated collecting system Absent uterus Hypospadias, cryptorchidism	33%
Gastrointestinal anomalies	Gastroesophageal reflux Esophageal atresia Impaired swallowing Hirschprung disease Imperforate anus	30%
Central nervous system anomalies	Cerebral atrophy Polymicrogyria Atrophy of the hippocampus Cerebellar atrophy	8%
Endocrine anomalies	Thyroid gland aplasia/hypoplasia Retrocarotid and retroesphageal thyroid extension Inhomogeneous thyroid structure Parathyroid gland dysfunction Growth hormone deficiency	65%
Skeletal and muscular anomalies	Cervical spine anomalies Thoracic vertebral anomalies Arachnodactyly, Camptodactyly, Syndactyly Hammer toes Skull malformations Diaphragmatic hernia	17–19%
Immune disorders	Athymia Thymic hypoplasia Ectopic thymus	75%

**Table 2 ijms-24-08317-t002:** Genetic modifiers influencing the *TBX1* penetrance and affecting the phenotypic expression in 22q11.2 DS.

Genetic Modifier	Role	Phenotypic Expression
*CRKL* (CRK like proto-oncogene adaptor protein)	Activates the RAS and JUN kinase signaling pathways, mediates transduction of intracellular signals	Development of organs originating from the neural crest, thymus, parathyroid glands, craniofacial structures, T lymphocytes, cardiac outflow region
*SLC2A3* aka *GLUT3* (Solute carrier family 2 member 3)	Facilitated glucose transporter	Conotruncal heart region, aortic arch
*KANSL1* (KAT8 regulatory NSL complex subunit 1)	Histone acetyltransferase complex member	Developmment of aortic arch, semilunar valve, cardiac septa, pulmonary artery
*JMJD1C* (jumonji domain containing 1C)	Chromatin expression modification, histone demethylation	Pharyngeal apparatus, cardiac outflow region
*RREB1* (Ras responsive element binding protein 1)	Chromatin expression modification, histone demethylation	Conotruncal heart region
*SEC24C* (SEC24 family member C)	Role in transporting proteins from the endoplasmic reticulum to the Golgi apparatus	Embryonic development, cardiac outflow region
*MINA* (MYC induced nuclear antigen)	Chromatin expression modification, histone demethylation	Cardiac development
*KDM7A* (Lysine-specific demethylase 7A)	Chromatin expression modification, histone demethylation	Cardiac development
*DGCR8* (DiGeorge syndrome critical region 8)	miRNAs and lnRNAs regulation	Embryo development, Immune, naurological, and cardiac functions
*NCOR2* (Nuclear receptor corepressor 2)	Transcriptional regulator of the retinoic acid signaling	B and T lymphocytes, regulation of the immune response
*EP300* (E1A binding protein P300)	Transcriptional regulator of the retinoic acid signaling	B and T lymphocytes, regulation of the immune response

**Table 3 ijms-24-08317-t003:** Autosomal recessive conditions resulting from mutations associated with 22q11.2 hemizygosity.

Gene	Disease	OMIM#	Phenotype
*PRODH*	Hyperprolinemia type 1	239500	General: neurological deficits Specific: psychomotor delay, hypotonia, seizures
*SLC25A1*	D2A2AD syndrome	615182	General: severe muscular weakness, respiratory distress, failed psychomotor development, early death Specific: encephalopathy, seizures
*GP1BB*	Bernard-Soulier syndrome	231200	Specific: hematologic disease, thrombocytopenia, increased megakaryocytes
*SCARF2*	Van den Ende-Gupta syndrome	600920	General: joint dislocations Specific: contractual arachnodactyly, hooked clavicles, blepharophimosis
*SNAP29*	CEDNIK syndrome	609528	General: neuropathy Specific: cerebral dysgenesis, ichtyosis, keratoderma

## Data Availability

Research data available by authors.

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
