# Peer review of "Chromosome 22q11.2 Deletion Syndrome: A Comprehensive Review of Molecular Genetics in the Context of Multidisciplinary Clinical Approach"

_ijms, 2023, doi:10.3390/ijms24098317_

Round 1

Reviewer 1 Report

This is a well-researched and up-to-date review of the features and genetics of 22q11.2 deletion syndrome.

A general comment is that the abstract mentions highlighting the molecular genetics "in correlation with a clinical multidisciplinary approach." The text seemed a little light on the "clinical multidisciplinary approach" part.

1.  Line 172. You correctly mention that VPI/hypernasal speech are common complications and that surgical intervention can be very helpful. I recommend adding a sentence that emphasizes that many new 22q Centers, many of which have surgeons who specialize in VPI surgery, have been opened/grown over the past decade. (This is definitely true in the US. You may have to investigate the degree to which this is true in Europe.)

2. Line 250. (An important distinction in clinical presentation), I would emphasize that hypocalcemia from hypoparathyroidism, if present, is more likely to present very young (infancy) whereas thyroid disease is more likely to present later (older childhood, early adulthood).

3. Line 381. "FISH is perceived as the gold standard genetic testing method" is not really true in the U.S. You could change this to "In the region where we practice medicine, FISH is perceived..." I would rewrite this section to note that microarray analysis is commonly used in clinical practice for first-line diagnostic evaluation. I would move line 388 to the end of the paragraph to mention that it is anticipated that whole genome sequencing may be the first-line approach in the not-too-distant future.

4. Line 414. It's not technically accurate to call CVS/amnio "combined screening." CVS/amnio are prenatal diagnostic tests.

5. Line 420. I believe that not every cell-free fetal DNA non-invasive prenatal screening test looks at 22q. I would research this more and adjust comments as necessary. In my experience, the microdeletion syndromes (including 22q) are sometimes offered as an opt-in check box for the patient undergoing the testing.

6. As your abstract mentions a clinical multidisciplinary approach as one of the aims of your paper, I do think it is appropriate for you to review PMID 32787583 "Impact of Interdisciplinary Team Care for children with 22q11.2 deletion syndrome" as there are actual measures of outcomes related to a multidisciplinary approach. 

Author Response

General comment

In the abstract, I have only mentioned the clinical symptomatology in 22q11.2 DS. In the text, I have added remarks on the role of multidisciplinary care in achieving better outcomes.

Ad 1.

Across many regions of the world, the number of centers of excellence with plastic surgery specialized in VPI surgery has grown over the past decade. I have added this information to the text. These centers may not be exclusively dedicated to 22q11.2 DS but surgical interventions are also indicated in velopharyngeal dysfunctions in other syndromic disorders.

Ad 2.

I have added a comment on an important distinction in the clinical presentation of hypocalcemia and hypothyroidism related to the patient’s age

Ad 3.

FISH remains a common diagnostic test for 22q11.2 DS with a high detection rate of approximately 95%. Certainly, an important limitation of this method is a failure to identify atypical and nested deletions as it can recognize deletions in the proximal part of the critical region, including the typical LCR22A-D deletion. In the Central European region, MLPA characterized by a higher detection rate and extent of the deletion is more widely used.

              I have rewritten the section according to your recommendation to highlight that microarray tests are first line methods and WGS is anticipated to be in the future.

Ad 4.

 I have corrected the sentence

Ad 5.

22q11.2 microdeletion is included in the NIPS panel (trisomies 21, 13, 18, aneuploidies XXX, XXY, XYY, X0, and 22q11.2)

Ad 6.

I have shortly extended important conclusions on the role of multidisciplinary team care. Relevant references have also been added

Reviewer 2 Report

1. In deletion negative patients, whether TBX1 gene mutation analysis is recommended? If so in what % of cases this gene is mutated.

2. would you  like to say that 22q 11.2 deletion is a  multisystem disorder not specifically confined  to only Di-George syndrome and Cono-truncal defects.

3.Discuss about its role  in intellectual disability  including Fragile X- Syndrome without cardiac defect. 

4. Is it possible to classify the phenotype based on the genes involved in the deleted region

5.Any explanation or work done on the common ocurrence of this De-novo mutation? Embryonic lethality of this deletion in aborted foetusses?

6. Role of Thymic transplant  in thymic hypoplasia bebore cardiac surgery?

Over all, the review appears to be educative

Author Response

Ad 1.

Chen et al. (Ultrasound Obstet Gynecol 2014; 43:396-403) showed an association of TBX1 variant with CTD in fetuses negative for 22q11.2 microdeletion on FISH. In 1 of 8 fetuses (12,5%), CGH array showed submicroscopic microdeletion in the TBX1 gene. Yagi et al. reported 3 mutations in two patients with VCF and DGS 9Lancet 2003; 362:1366-1373). Furthermore, a novel loss-of-function mutation was identified by Xu et al. (BMC Med Genet 2014; 15:78) in one out of 199 patients with isolated CTD and no 22q11.2 deletion. Subsequently, case reports describing patients with phenotypes corresponding with 22q11.2 deletion but negative FISH were provided by Haddad et al. (Clin Diabetes Endocrinol 2019; 5:13). TBX1 gene mutations in patients with high clinical suspicion of VCF/DGS has been shown in small groups. However, referring to the role of TBX1 in 22q11.2 DS, in patients with clinically evident disease in whom FISH or microarray tests are not confirmatory, TBX1 gene testing is recommended. Relevant references have been added to the reference list

Ad 2.

22q11.2 DS is characterized by a high degree of phenotypic variablity, not only including VPI, CTD, and hypoplastic thymus. Expanded clinical symptomatology including, but not limited to endocrine problems, psychiatric disease, immune dysregulation, and developmental anomalies make 22q11.2 a multisystem disorder

Ad 3.

High prevalence of somatic complaints, thought and attention problems are observed in children and adolescents with either 22q11.2 DS or Fragile X syndrome. These syndromes are not always associated with intellectual disability, potentially allowing affected patients to express their thoughts and feelings more clearly yet syndrome-specific contributions appear to occur in different types of psychiatric illness (Glasson EJ et al. J Am Acad Child Adolesc Psychiatry 2020; 59:1036-1048)

Ad 4.

I believe it is possible but has not been applied to practice

Ad 5.

22q11.2 DS results mainly (in 90-95% of newly identified cases) from de novo non-homologous meiotic recombination. It has been identified in as many as 1:100 fetuses with major structural abnormalities such as CHD and approximately 1:1000 seemingly anatomically normal fetuses  pointing to embryonic lethality (McDonald-McGinn DM et al. Nat Rev Dis Primers 2015; 1:15071)

Ad 6.

To facilitate the optimal establishment of thymic allograft, stability of comorbidities, such as cardiopulmonary function, upper airway stabilization, appropriate weight gain, and metabolic compensation. The management of concurrent disorders plays a fundamental role in the timing of thymus transplantation (Howley E, et al. Ther Clin Risk Manag 2023; 19:239-254)

Round 2

Reviewer 2 Report

All queries asked were satisfactorily tackled

The auth